# Donkey Slaughter in Brazil: A Regulated Production System or Extractive Model?

**DOI:** 10.3390/ani15111529

**Published:** 2025-05-23

**Authors:** Sharacely de Souza Farias, Aline Rocha Silva, Rayane Caroline Medeiros do Nascimento, Marisol Parada Sarmiento, Tobyas Maia de Albuquerque Mariz, Pierre Barnabé Escodro

**Affiliations:** 1Department of Veterinary Medicine, Campus of Engineering and Agricultural Sciences (CECA), Federal University of Alagoas, Maceio 57200-000, Brazil; aline.silva@ceca.ufal.br (A.R.S.); rayane.nascimento@ceca.ufal.br (R.C.M.d.N.); pierre.escodro@vicosa.ufal.br (P.B.E.); 2Department of Preventive Veterinary Medicine and Animal Health, School of Veterinary Medicine and Animal Science, University of São Paulo, São Paulo 05508-220, Brazil; mparadasarmiento@unite.it; 3Department of Animal Science, Campus of Engineering and Agricultural Sciences (CECA), Federal University of Alagoas, Maceio 57200-000, Brazil; tobyas.mariz@arapiraca.ufal.br

**Keywords:** donkey, *Equus asinus*, animal welfare, animal extraction

## Abstract

Donkeys were once important working animals in Brazil, but their role has declined with mechanization. Today, they are mainly slaughtered for their skins, which are exported to produce ejiao, a gelatin used in traditional Chinese medicine. However, donkey slaughter in Brazil is not properly regulated, raising concerns about animal welfare and sustainability. Complaints filed by public agencies in Brazil have raised concerns about the donkey slaughter chain in the country, questioning whether this production system operates under regulated standards or follows an extractive model. The objective of this study was to evaluate the production system of donkeys destined for slaughter in Brazil through physical and hematological assessments, aiming to identify potential systemic failures that may compromise animal welfare. This study evaluated 104 abandoned donkeys intended for slaughter using physical and blood tests to assess their health and identify potential welfare issues. For this, we evaluated the sex, estimated the age, assessed the body condition score (BCS), and performed hematological assessments, including the measurement of hematocrit (HT), total plasma protein (TPP), and plasma concentration of fibrinogen. Most animals showed signs of systemic inflammation, suggesting that they were not in ideal health. The findings highlight the risks of an extractive, poorly regulated system that does not ensure proper care or protection for these animals.

## 1. Introduction

The global donkey population is estimated at 50,453,888 individuals, including 395,910 in Europe and 2,843,306 in South America. In Brazil, the donkey population is estimated at 376,874 individuals, the majority of which (326,569 individuals—86.7%) are found in the Northeast region [1].

Donkey breeding in Brazil has evolved with different regional patterns. In the South and Southeast, donkeys are bred mainly for mule production [2]. In contrast, in the Northeast, donkeys are mainly used as draft animals by small-scale farmers and riverside communities, typically without the support of structured breeding systems [3].

Since 2016, the Brazilian Institute of Geography and Statistics has not conducted a population survey, resulting in a lack of reliable data, administrative records, and an accurate understanding of the actual donkey population in Brazil. The significant decline in the use of these animals on rural properties may have contributed to this lack of interest. In Brazil, the mechanization of agriculture and the replacement of donkeys with motorized vehicles [4] have significantly reduced the utility of donkeys as working animals over the past two decades [5].

The exclusion of donkeys from the global economy may be contributing to their decline, placing the species at serious risk of extinction [6,7], as evidenced by the decrease in their population, from 1.31 million in 1994 to 822,000 in 2018, representing a 37.2% decrease. The increasing demand for donkey skin imports, driven by the traditional Chinese medicine industry for ejiao production, may have further contributed to this decline [5,8].

Ejiao is believed to have therapeutic properties related to blood nourishment, immune response optimization, metabolic balance improvements, anti-aging effects, and the treatment of gynecological diseases [9]. Approximately 90% of the ejiao produced is imported from countries that lack established production chains, such as most African countries and South America [10]. In these countries, the capture, transport, holding, and slaughter of donkeys are often conducted improperly [11,12]. An organized production chain requires a standardized process, including raw material sourcing, pre- and post-slaughter handling, carcass classification and grading, refrigeration, packing, and transportation [13]. All these procedures must comply with official regulations and be carried out by qualified professionals who follow technical standards to ensure the integrity of the quality system [14,15].

In Brazil, the slaughter of equids, including donkeys, is legal under Decree No. 9013/2017, which regulates the inspection of animal products [16]. This law allows the processing of donkeys for meat, hides, and other products for domestic and export markets. Brazil does not have a specialized industry for donkey slaughter, which has led to the use of facilities and practices originally designed for other species. This mismatch has serious consequences for animal welfare. Pre-slaughter water baths, commonly used in cattle processing, are highly stressful for donkeys, often causing them to resist movement and necessitating the use of electric prods. Furthermore, the absence of a proper screening process to identify animals unfit for slaughter has led to the presence of pregnant females on the slaughter line, with some giving birth just moments before being sent to slaughter [5].

However, in the state of Bahia, there is a legal slaughterhouse for donkeys. In 2018, a court injunction was issued suspending the slaughter of donkeys for a period of 10 months, following investigations by the Civil Police and the Bahia Agricultural Defense Agency. These authorities registered 8 cases of glanders and 5 cases of equine infectious anemia at a triage farm in the city of Canudos. Additionally, numerous instances of mistreatment were reported, as donkeys were deprived of basic needs, such as water, food, and veterinary care [17].

In 2022, Brazil enacted the Self-Control Law (No. 14,515/2022), which shifted inspection responsibilities from government authorities to private agents operating within slaughterhouses [18]. While the goal was to streamline operations, the change raised significant concerns about increased risks to animal welfare and food safety due to reduced governmental oversight. Donkey meat is not consumed by the domestic market in Brazil, with hides being the only product derived from donkey slaughter, primarily used for the production of ejiao [19].

The slaughter of donkeys in Brazil for skin export has profound long-term consequences, including a significant reduction in genetic diversity and an imminent risk of species extinction [20,21]. One way to assess animal welfare is by conducting physical, hematological, and hormonal evaluations. Physical assessments provide insights into the animals’ nutritional and health status, while hematological assessments offer additional, complementary information. Among the most important hematological parameters are fibrinogen and total plasma protein levels, which often change in response to injury, trauma, or disease. These markers are therefore valuable indicators of an animal’s overall health and welfare. The objective of this study was to evaluate the production system of donkeys destined for slaughter in Brazil through physical and hematological assessments and to determine whether systemic failures exist that compromise animal welfare.

## 2. Materials and Methods

### 2.1. Ethics Approval

This study was approved by the Committee for the Use and Care of Animals in Research (CEUA) of the Federal University of Alagoas (UFAL), Brazil, under registration number 21/2019.

### 2.2. Study Site and Period and Animals

The experiment was conducted at Fazenda Santa Isabel and Fazenda Piedade, in the city of Canudos, State of Bahia, Brazil, in 2019. The climatic subtype of the region is semi-arid, according to the Köppen–Geiger climate classification, characterized by high temperatures throughout the year.

The experiment was conducted from February to April 2016. During this time, an assessment of the health and animal welfare of 104 abandoned donkeys (no specific breed or age defined) was conducted. These animals were selected from a group of 300 abandoned donkeys available for the study. Data collection was limited to 104 animals due to their high reactivity, stress, and lack of habituation to handling. This also affected the number of animals from which blood samples were collected.

The assessment was conducted through the collection of data on physical and hematological evaluations during technical visits, including age, sex, body score, neck condition score, hematocrit, globular volume mean corpuscular volume, total plasma protein, and fibrinogen.

#### 2.2.1. Physical Variables

Age was assessed through the dental arch examination, observing the eruption of the incisor teeth and the appearance of the dental star of the permanent incisor teeth [22] and sex.

Body score: This evaluation was used on a scale from 1 (extremely thin) to 9 (obese) [23]. This body score system determines the condition based on the general physical appearance, bone tuberosities, and rib coverage.

The neck condition score: This evaluation was based on a scale of 0 (thin neck) to 4 (obese) [24].

#### 2.2.2. Hematological Variables

Blood samples from each animal were collected from the jugular vein in a vacuum tube system containing EDTA 10% and 1 G needles (Vacuntainer^®^). The samples were refrigerated at 8 °C and transported immediately to the laboratory for analysis. The values determined were HT, VG, PPT, and fibrinogen.

HT measurement was performed using a microhematocrit centrifuge with 75 mm capillary tubes. The samples were centrifuged at 10,000 rpm for 5 min and subsequently analyzed using a microhematocrit reader [25]. Fibrinogen was carried out using the precipitation method at 56° C in microhematocrit tubes by calculating the difference between plasma and serum protein concentration [26].

A blood sample was collected from only 88 out of 104 animals due to their high reactivity and lack of familiarity with routine handling, which made it difficult to collect the animals.

The HT or GV was measured to detect possible anemia, with a minimum healthy threshold of 28% [27,28]. TPP was measured to identify possible dehydration, with values above 8 g/dL indicating dehydration [29]. Plasma fibrinogen concentration was analyzed to determine the degree of systemic inflammation, as it is a positive regulatory acute-phase protein, with a tolerable value of up to 400 mg/dL [30].

### 2.3. Statistical Analysis

Descriptive statistics were performed with all variables, with the results always stratified by sex. A Shapiro–Wilk test was performed to test the normality of the variables. Spearman correlations were calculated for all variables, with the *p*-value < 0.05 considered significant. All analyses were conducted using the programming language R Verson: 2024.12.1+563 [31]. The graphics were completed using the package ggplot2 Version 3.5.2 [32].

## 3. Results

The total population of animals (*n* = 104) included data on females, entire males, and castrated males (Figure 1), body condition score, neck condition score, and age. Hematological variables were obtained from a subset of the population (*n* = 88). A detailed description of all variables stratified by sex categories is presented in Table 1.

The majority of abandoned donkeys (*n* = 50–48.08%) were entire males, of which 46 had hematological data variables. The median age of all animals was 7 years (*n* = 104; SD 8.18; minimum 0.8; maximum 40; Shapiro–Wilk test *p*-value < 0.001).

The donkeys were grouped into 5-year intervals and stratified by category: female, entire male, and castrated male (Table 2). Among the 104 animals evaluated, the majority of those in the 0- to 5-year age group were females (*n* = 23, 50%).

The body condition score and neck condition score data are described in Table 3 and Table 4, stratified by different categories and including the median age for each score. Based on Table 3, the majority of females and castrated males presented a body condition score of 3 and 4 (females *n* = 25, 54.3%; castrated males *n* = 5, 62.5%), whereas the majority of entire males had a body condition score of 2 or 3 (*n* = 36, 52%).

Concerning the neck condition score (Table 4), the majority of females had a score of 2 (*n* = 21, 45.6%), while the predominant score among entire males and castrated males was 1 (males: *n* = 22, 44%; castrated males: *n* = 3, 37.5%).

The hematological data are graphically presented in Figure 2, Figure 3 and Figure 4, with the variables stratified by different categories. The majority of animals had hematocrit levels above 25% (Figure 2), total plasma protein levels below 7.5 g/dL (Figure 3), and fibrinogen levels exceeding 400 mg/dL (Figure 4). The box plots for all the hematological variables showed overlapping distributions between categories, suggesting that there are likely no significant category-based differences.

The correlation analyses identified three positive correlations: between body condition score (BCS), neck condition score (NCS), and age. BCS was strongly and positively correlated with NCS (Spearman rho = 0.755; *p*-value < 0.001). Age showed a weak positive correlation with BCS (Spearman rho = 0.290; *p*-value < 0.01) and NCS (Spearman rho = 0.323; *p*-value < 0.001). No other significant correlations were found. The results of all correlation analyses are presented in Table 5.

## 4. Discussion

In this study, we systematically compared a group of abandoned donkeys rescued from slaughter in Bahia and identified a predominance of males (entire and castrated) over females, which contrasts with findings from other studies [33]. However, the hematological results were consistent with previous research, showing no significant differences in hematological values between these categories [34,35].

Since the prior management history of these animals was unknown, age estimation was performed using a dental arch examination [22,36]. This assessment was also crucial for identifying dental changes, which may be associated with the breeding system and the type of diet these animals received [37]. In ranging conditions, tooth wear is proportional to tooth eruption due to the silica present in the pasture. However, when an animal experiences forage limitations or lacks access to pasture, restricted wear can lead to the development of pronounced transverse ridges [38].

The working capacity of draft horses is influenced by numerous factors that predispose them to various illnesses, leading to behavioral, physiological, and clinical changes. The main factors include climate, management, age, body condition score, diet, working hours, rest, and water availability [39,40]. After the BCS assessment, it was observed that most of the analyzed animals had a body condition below the healthy threshold. As previously mentioned, a poor body condition score reduces work capacity and increases susceptibility to disease, reflecting neglect in maintaining essential welfare indicators.

The BCS results were confirmed with the NCS, as most animals were underweight, with over 80% scoring between 0 and 2. A study in Patos-PB evaluated horse health using BCS and hematological analysis of 36 horses. Among the assessed animals, 47% had a poor body condition score [40]. This low BCS was attributed to the intensity of work, along with fluctuations in food and water availability during the dry season. These conditions were linked to overwork, mistreatment, and inadequate food and water supply, as well as complete abandonment, which ultimately motivated this research.

According to the data obtained in the HT variable, most animals did not present with dehydration and/or anemia. Factors such as age, sex, breed, and physical exercise can influence hematological values. For instance, horses under stress or fear, for example, may exhibit elevated hematocrit due to the large reserve of red blood cells in the spleen and their rapid release into circulation [41]. Similar physiological responses can also affect blood test results in donkeys. Therefore, it is essential to keep the animals calm during blood collection to ensure accurate measurements [27,42].

Our study showed HT values within a range of 24.3% to 39.6%, which is normal for donkeys [43]. The values in this study were above 25%. This can be explained by animal physiology because donkeys have a high capacity to preserve circulating plasma volume when subjected to dehydration, with no direct correlation between hematocrit levels and the degree of dehydration. Additionally, donkeys experience a hematocrit increase when exposed to high altitudes for three weeks or less, making it challenging to assess their hydration status based solely on laboratory results [44,45]. On the other hand, in the TPP evaluation, the majority of animals (90.27%) showed values below 7.5 g/dL, and the species-specific reference range was between 5.84 and 6.93 g/dL [46]. TPP values are generally used to assess dehydration, which could be the case in this study. TPP reflects the overall physiological state: low levels may indicate malnutrition or parasitic infections, while elevated values, especially when associated with high hematocrit, suggest dehydration [47]. Additionally, plasma proteins are primarily produced by the liver and serve as indicators of nutritional status and liver function. Hyperproteinemia is associated with dehydration, while hypoproteinemia is linked to liver dysfunction [42,48]. Regarding plasma fibrinogen concentration, all animals exhibited fibrinogen levels exceeding the limits established as acceptable in the literature, demonstrating systemic inflammation, infection, or stress conditions, making it a useful indicator of animal health and welfare [49]. As a positive acute-phase protein, fibrinogen plays a regulatory role in the inflammatory response, with a tolerable threshold of up to 400 mg/dL [30,50]. Normal fibrinogen and TPP levels indicate good health and animal welfare. The physical and variables combined with the environmental conditions of the donkeys indicated signs of malnutrition and mistreatment of most animals.

The positive correlation observed among the selected parameters confirms their effectiveness in assessing animal health. However, as demonstrated in other studies, the donkeys housed on the referenced farm were in poor welfare conditions [17]. It is well known that debilitated animals tend to isolate themselves from the herd, and it is possible that the measurements were carried out on healthier animals, as they were more docile and easier to handle during sample collection [51].

Between 1966 and 2016, Brazil experienced a 37.08% reduction in its donkey population, driven by the increasing demand for fur trade [12]. This finding is in line with other reports highlighting the intensification of the slaughter of donkeys in the Northeast region in recent years. One study highlighted that a single slaughterhouse was responsible for exporting 44,000 donkey skins, accounting for approximately 25% of the country’s annual target for 2018 [52].

Furthermore, these animals can be collected from roads or abandoned farms at no cost, without sanitary control or transport regulations [21]. Thus, in addition to these economic facilities, the exploitative trade of donkeys, driven solely by profit, neglects their health and welfare, as widely documented in the literature and further confirmed by our study.

This economic justification does not support the unsustainable extractive activity observed in our study of these animals. Donkeys endure suffering at every stage of the fur trade, from capture and transport to slaughter [53,54]. Moreover, untrained workers often use inhumane and illegal methods to kill these animals, causing severe pain, fear, and distress.

Furthermore, these animals can suffer both physical and psychological distress when separated from their companions or transported, often resulting in illness or disability [54,55]. The abandonment and poor conditions in which the animals in this study were found are consistent with all the suffering this species has been subjected to over the last few years in the country, particularly in the Northeast region, due to neglect and uncontrollable fur commerce.

It is undeniable that the welfare and survival of donkeys are increasingly threatened as the demand for their skin rises. In Brazil, where no structured reproductive chain or established production system for donkeys exists, the economic feasibility of such production remains largely unknown. The unregulated and indiscriminate slaughter of these animals places them at imminent risk of extinction [56].

Given the absence of a structured reproductive chain and the growing threat to donkey welfare and survival due to the increasing demand for their skin, it is imperative to critically assess whether the slaughter of donkeys is truly necessary or justifiable. Although legally permitted in Brazil, this practice remains economically unviable and ethically controversial, especially in a country with no cultural tradition of consuming donkey meat. If the activity is to persist, it must be restructured under clear, species-specific regulations, enforced through effective oversight and grounded in public policies that prioritize environmental conservation and animal welfare. Ultimately, any decision regarding the future of this practice must be guided by ethical considerations, scientific evidence, and a commitment to preventing irreversible harm to both animals and biodiversity.

## 5. Conclusions

Based on the data analysis, the animals were characterized as being in poor health. Most of the donkeys exhibited altered body and neck condition score, plasma fibrinogen concentration, and total plasma protein, indicating systemic inflammation, despite the other laboratory parameters showing no signs of disease. However, physical assessments revealed evidence of abandonment, malnutrition, and mistreatment, highlighting the poor welfare conditions affecting these donkeys. The findings of our study, combined with the growing demand for donkey skin and the widespread negligence and cruelty toward these animals, underscore the urgent need to establish stricter laws and regulations to control and regulate the escalating slaughter. Additionally, the implementation of public policies that prioritize animal health and welfare is essential to ensuring comprehensive protection for donkeys across all sectors. Finally, considering the conditions of the animals found in our research and based on the current reality of the species, which is threatened by neglect and the ambitions of the fur trade, it is evident that no well-established donkey production chain exists in Brazil. Any activity dominated by animal abuse constitutes an unsustainable extractive practice, ultimately leading to the species’ extinction. Therefore, an increase in research on donkeys is necessary, given the severity of the issue and the scarcity of extensive studies in the scientific literature. This study aims to serve as a reference for discussions on the intersection of animal welfare, economic interests, and environmental concerns related to donkeys, particularly in Brazil’s northeast region.

## Figures and Tables

**Figure 1 animals-15-01529-f001:**
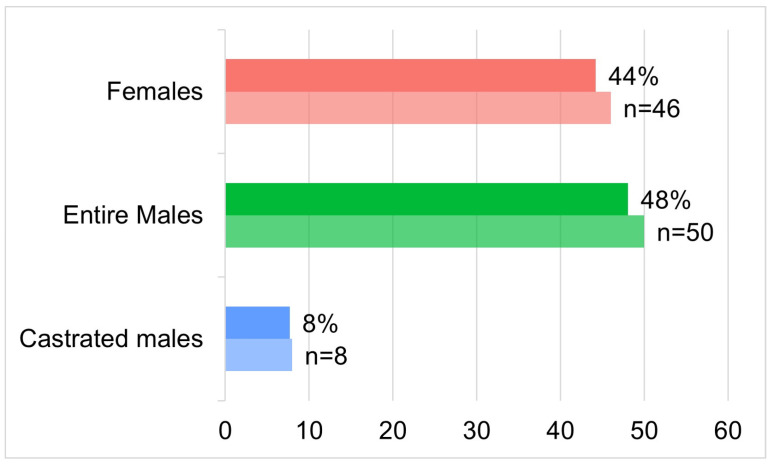
Percentage and number of abandoned donkeys (*n* = 104) distributed into three categories: females, entire males, and castrated males.

**Figure 2 animals-15-01529-f002:**
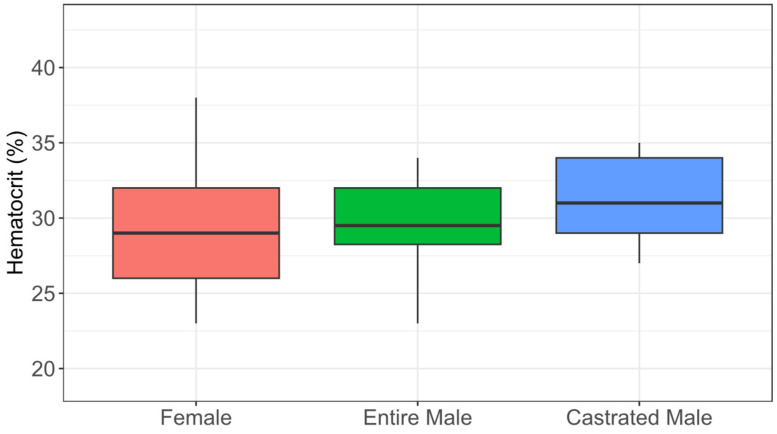
Hematocrit of abandoned donkeys by the female (F), entire male (M), and castrated male (Mc) categories represented in a box plot.

**Figure 3 animals-15-01529-f003:**
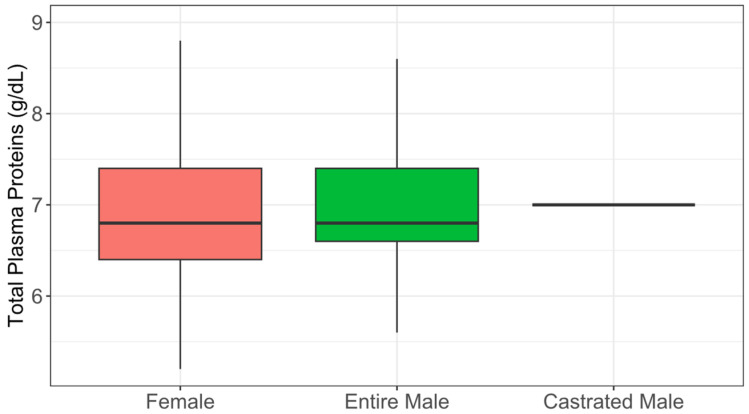
Total plasma protein of abandoned donkeys by the female (F), entire male (M), and castrated male (Mc) categories represented in a box plot.

**Figure 4 animals-15-01529-f004:**
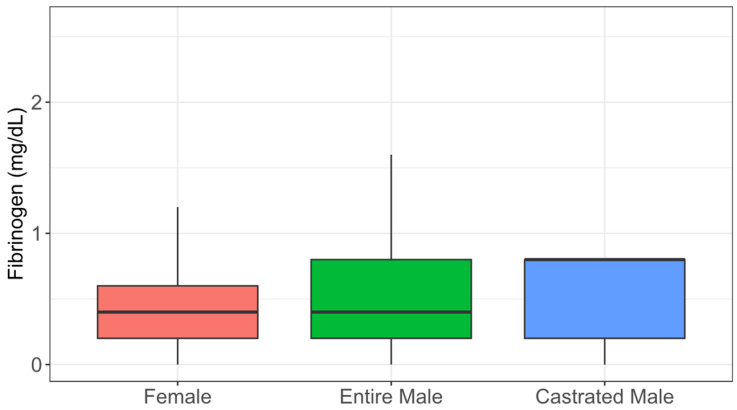
Total fibrinogen of abandoned donkeys by the female (F), entire male (M), and castrated male (Mc) categories represented in a box plot.

**Table 1 animals-15-01529-t001:** Descriptive statistics of physical (*n* = 104) and hematological (*n* = 88) evaluation of abandoned donkey distributed into three categories: females, entire males, and castrated males. N: Number of individuals; SD: standard deviation; Min: minimum value; Max: maximum value; SWilk: Shapiro–Wilk *p*-value.

Parameters	Female						Entire Male						Castrated Male					
	N	Median	SD	Min	Max	SWilk	N	Median	SD	Min	Max	SWilk	N	Median	SD	Min	Max	SWilk
Body condition score	46	3.0	1.4	1.0	6.0	0.01	50	3.0	1.2	1.0	7.0	<0.001	8	4.0	1.2	2.0	6.0	0.792
Neck condition score	46	2.0	0.9	0.0	4.0	<0.001	50	2.0	0.8	1.0	4.0	<0.001	8	2.0	1.1	1.0	4.0	0.197
Age (years)	46	5.5	6.4	1.6	25.0	<0.001	50	9.0	9.3	0.8	40.0	<0.001	8	9.5	8.3	3.0	27.0	0.365
Hematocrit (%)	37	29.0	3.8	23.0	38.0	0.338	46	29.5	4.1	19.0	43.0	0.001	5	31.0	3.3	27.0	35.0	0.737
Total protein (g/dL)	37	6.8	0.7	5.2	9.0	0.027	46	6.8	0.7	5.6	8.8	0.094	5	7.0	0.8	5.2	7.0	<0.001
Fibrinogen (mg/dL)	37	0.4	0.5	0.0	2.2	<0.001	46	0.4	0.7	0.0	2.6	<0.001	5	0.8	0.9	0.0	2.2	0.277

**Table 2 animals-15-01529-t002:** Age percentage of abandoned donkeys (*n* = 104) in three different categories: castrated males, entire males, and females.

	Female		Entire Male		Castrated Male		Total	
Age Range (Years)	N	%	N	%	N	%	N	%
0–5	23	50	18	36	2	25	43	41.4
6–10	11	23.9	11	22	2	25	24	23.1
11–15	5	10.9	6	12	1	12.5	12	11.5
16–20	4	8.7	6	12	2	25	12	11.5
>20	3	6.5	9	18	1	12.5	13	12.5
Total	46	100	50	100	8	100	104	100

N: Number of individuals; %: number of individuals expressed in percentage.

**Table 3 animals-15-01529-t003:** Median values and percentage of body condition scores of three different categories of abandoned donkeys: castrated males, entire males, and females.

	Female			Entire Male			Castrated Male		
Score	N	%	Median Age (Years)	N	%	Median Age (Years)	N	%	Median Age (Years)
1	4	8.7	2.3	2	4	9.5	0	0	-
2	8	17.4	4.0	16	32	4.3	1	12.5	7.0
3	13	28.2	6.0	20	40	12.5	2	25	3.0
4	12	26.1	7.0	6	12	7.0	3	37.5	17.0
5	4	8.7	11.0	3	6	3.6	1	12.5	17.0
6	5	10.9	8.0	2	4	23.5	1	12.5	8.0
7	0	0	-	1	2	21.0	0	0	-
8	0	0	-	0	0	-	0	0	-
9	0	0	-	0	0	-	0	0	-
Total	46	100	-	50	100	-	8	100	-

N: Number of individuals; %: number of individuals expressed in percentage.

**Table 4 animals-15-01529-t004:** Median values and percentage of neck condition score as a prompt of body condition of three different categories of abandoned donkeys: castrated males, entire males, and females.

	Female			Entire Male			Castrated Male		
Score	N	%	Median Age (Years)	N	%	Median Age (Years)	N	%	Median Age (Years)
0	3	6.5	2.6	0	0	-	0	0	-
1	12	26.1	4.0	22	44	6.0	3	37.5	7.0
2	21	45.6	6.0	17	34	8.0	2	25	7.0
3	9	19.6	12.0	10	20	16.0	2	25	12.5
4	1	2.2	8.0	1	2	20.0	1	12.5	27.0
Total	46	100	-	50		-	8	100	-

N: Number.

**Table 5 animals-15-01529-t005:** Correlation matrix (Spearman method) between physical (*n* = 104) and hematological (*n* = 88) variables of abandoned donkeys.

		BCS	NCS	Age	Hematocrit	Total Protein	Fibrinogen
BCS	Spearman’s rho	-					
	Df	-					
	*p*-value	-					
NCS	Spearman’s rho	0.755 ***	-				
	Df	102	-				
	*p*-value	<0.001	-				
Age	Spearman’s rho	0.290 **	0.323 ***	-			
	df	102	102	-			
	*p*-value	0.003	<0.001	-			
Hematocrit	Spearman’s rho	0.020	−0.030	0.154	-		
	Df	86	86	86	-		
	*p*-value	0.850	0.781	0.152	-		
Total protein	Spearman’s rho	0.141	0.074	0.186	0.080	-	
	Df	86	86	86	86	-	
	*p*-value	0.190	0.494	0.084	0.457	-	
Fibrinogen	Spearman’s rho	−0.111	−0.064	−0.133	−0.198	0.161	-
	Df	86	86	86	86	86	-
	*p*-value	0.305	0.551	0.217	0.065	0.134	-

Note: ** *p* < 0.01, *** *p* < 0.001.

## Data Availability

Publicly available datasets were analyzed in this study. This data can be found here: https://data.mendeley.com/datasets/7n9t5fn99b/2 (accessed on 13 January 2025).

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
