# Peer review of "Donkey Slaughter in Brazil: A Regulated Production System or Extractive Model?"

_animals, 2025, doi:10.3390/ani15111529_

Round 1
Reviewer 1 Report
Comments and Suggestions for Authors
Overall, a very interesting paper. Please change donkey fur to donkey skin or donkey skin trade industry. Also, in the results and discussion, can you describe in more details the difference in the on site field testing and lab testing and why there was a difference? could it be lab values compared to horse standards or storage or delayed time to lab? Also, why were the values chosen as indicators of welfare (e.g. fibrinogen, total plasma protein) I think a better argument should be presented. Also, why was disease testing not conducted? It was mentioned in the intro that EIA and Glanders were found in donkeys in Bahia at the slaughter facility, could these donkeys have been exposed or tested positive. Describe conditions of how the donkeys were kept and or abandoned, I didn't understand that part. Overall, very important work and we need to know more about the conditions of donkeys sent to slaughter for the skin trade industry and consider methods to improve their welfare. Thank you for researching this important topic.
Author Response
Caro Revisor, Gostaria de agradecer sinceramente por todos os seus comentários. O artigo foi melhor estruturado e todas as respostas às suas sugestões podem ser encontradas na tabela anexa.

Reviewer 2 Report
Comments and Suggestions for Authors
The study appears to be a highly interesting and necessary document of denunciation, highlighting significant shortcomings in the monitoring and adequate management of the welfare and conservation of the donkey species in Brazil. However, in the reviewer's opinion, it cannot be published in its present form.
General consideration and main shortcomings:
TITLE AND STUDY AIM
Both the title and the aim in the present form appear not clear and not fully centred with the objective of the study, which appears to focus on the lack of adequate donkey welfare management on the farm and during transport, denouncing a system that does not protect animals to any degree, notwithstanding the slaughter system and its regulation.
INTRODUCTION
- The introduction should include a general presentation of donkey farming evolution in Brazil (e.g. regional differences, local extensive production)
- The introduction does not include any reference to the current Brazilian legislation on food safety, slaughter supervision, meat hygiene and animal welfare. Given the repeated emphasis on the 'regulated system' in the manuscript, it is essential to provide an overview of the regulatory framework in this area.
In terms of slaughter management, in particular, it would be beneficial to briefly present the main changes introduced, starting from 2022, with the enactment of the Self-control Law.
- It would be beneficial to incorporate a concise description of the utilisation of donkey-derived products within the slaughter chain, along with the export-oriented production of donkey skin,
RESULTS AND DISCUSSION
- A clear mention of the criteria and rationale that were employed in the selection of the zootechnical and clinical parameters that were incorporated into the study is lacking.
Pointed notes:
Lines 55-56 for completeness, please include a brief and explicit definition of Eijao.
Line 63-66: Please see the general comment regarding the lack of a clear regulatory framework for the authorisation and management of slaughter at a national level.
Lines 87-88: “These visits focused on a group of 104 abandoned donkeys, which had been victims of slaughter and export of donkey skins from Brazil to China.”
The paragraph needs to be rewritten, as it suggests that the animals were slaughtered to be destined for skin export, whereas the previous paragraph states that they were legally seized under judicial supervision thanks to the intervention of the National Forum for the Protection and Defence of Animals in the city of Canudos-BA.
Lines 123-124: Why were the haematological variables obtained from a subpopulation? No explanation of this choice is given in any section
Lines 194-195: The authors posit that only 50 of 104 donkeys were abandoned. Therefore, the question arises as to why it was impossible to find any prior management history for any of the animals. This particular issue is not addressed in the subsequent discussion. Given that the slaughterhouse from which the animals were retrieved was duly authorised, should some form of traceability not have been maintained, at least for the animals not classified as 'abandoned'? This aspect remains unaddressed.
Lines 223-224: Is a reference missing? Which physiological ranges/data do the authors refer to when they define 'expected haematological data for abandoned donkeys'?
Comments on the Quality of English LanguageThe manuscript would benefit from a revision by a native English speaker to enhance its fluency and readability.
Author Response
Dear Reviewer, I would like to sincerely thank you for all your comments. The article has been better structured, and all the responses to your suggestions can be found in the attached table.

Reviewer 3 Report
Comments and Suggestions for Authors
Characterization of Donkeys Withdraw From Slaughter in Bahia: Production Chain or Extrativism?
March 2025
General comments
Title needs rewriting - withdraw should be withdrawn. Currently I am unclear from reading the title and simple summary what the study is about, so needs rewriting to reflect the study.
Simple summary needs rewriting and expanding - this should reflect the main study/abstract but just be written in more simple terms avoiding any technical language (e.g. need to simplify or explain the terms ‘regulated production system’ (and remove the extra comma in this), and ‘extractive model’. Again, I am unclear from reading this what the study is about. It should be structured similar to the abstract – brief background / justification, primary aim, study design/methods, then results and conclusions.
Line 14 and throughout – where you are talking about multiple animals please use ‘were’ not ‘was’
Abstract
Line 23 – I don’t agree that donkeys are close to extinction, please remove or rephrase this.
Line 25-28 – needs more methods details. I suggest you read and follow the STROBE guidelines on observational studies (https://www.strobe-statement.org/) here and throughout, as the methods including the study population and sampling methods are unclear.
Line 28 – this is the first time you mention abandoned donkeys, so you need to provide details of what population you are looking at, how you sampled them, and how the clinical data was obtained, and how this relates to the slaughter chain.
Line 28 ‘was’ should be ‘were’ – please check for similar errors throughout the paper
Line 28-29 please state the units (e.g. years for age) and the body condition score system used otherwise the numbers are meaningless.
Line 29-30 please state the normal limits used as this varies.
The results would be better presented as mean +/- SD or median and range, depending on the data distribution.
Line 31-32 don’t make any sense when reading, as you have not explained how the data was obtained, e.g. what is lab and field analysis.
Line 33-37 – needs rewriting. This does not relate to a fur trade, you have not explained your study design in here, so I can’t assess whether or where there is mistreatment. I am unclear how this relates to slaughter, and no data has been presented on the production chain. As above, I think donkey extinction is unlikely considering the current numbers worldwide, so this needs rephrasing. You need to provide evidence of ‘rampant and cruel’ activities to justify this or consider rewording.
Introduction
As a whole, this is well written, and clears sets out the background for this study. This really needs explaining better in the abstract and summary to reflect this.
Line 48-50. I don’t understand how ‘government initiatives to facilitate access to agricultural inputs’ contributes to a decline – please explain.
Line 50-52. I don’t believe that donkeys are at risk of global extinction – I agree there has been a significant decline, please rephase this here and throughout.
Line 74 – please explain what an extractive model is
Line 74 – please remove the comma between production, system – I believe this is an error
Materials and methods
Please reorganise using the STROBE guidelines, so it is clear what the study design, setting population is, and then describe how it was sampled and measured.
Line 80 – what do you mean by ‘zootechnical’ records?
Line 81 – please provide details of the states
Line 87-88 please reword, if the donkeys had been victims of slaughter and export, then they would have been dead and left the country, so not available for clinical assessment
Line 85-89 – I am unclear whether the dates described in line 79 relates to the clinical examination, or whether data was collected by survey during these dates, and the clinical examinations were done at other times – please rewrite and clarify.
I am unclear at which stage the donkeys were examined - when seized in the states, or were they examined at a centre in Canudos-BA? I presume the latter, but then please provide information on how long after seizure they were examined.
Neck condition scoring – please shorten this section – if you are using a previously published system, then you just need to reference this and explain the range (0 (thin)- 4 (obese)
Line 103 – here and in abstract, please explain what you mean by Globular volume. Standard terminology would be mean corpuscular volume (which is different to haematocrit) or packed cell volume (which is similar to haematocrit but measured differently). Please check and define what measurement you are using. There is a helpful description on the Cornell website: https://eclinpath.com/hematology/tests/hematocrit/
Line 117 – additional information is needed in this section. Please see the STROBE guidelines, and add details on how you determined study size, how normality of data was assessed, and how descriptive data was handled (e.g. mean, median, mode)
See STROBE item 9, please also include information about how bias was managed
Results
Line 123-124 – please provide further details of why only a subset had blood samples, and how these were selected.
Line 127 and throughout where applicable, please provide SD or SEM in addition to the mean in the text, and the number of animals that this data was available for, e.g. mean +/- SD (n=xx). Please confirm in methods whether the data was tested and had a normal distribution, and therefore whether mean was the most appropriate measure.
Legends for tables and figures need to be stand alone from the text. This means that they should include sufficient information about the study design to be read alone without the text. Please add details about the study design and population to the titles to achieve this.
Please change the Male category to Male entire, as Male castrated are also Male.
Line 136-137 – method of ageing should have been in methods not results. Please provide further details as this is known to be an inaccurate approach, so please describe the system you are using, e.g. incisor eruption, incisor wear etc, and make it clear that this was used as an estimate not an accurate methods of ageing.
Figure 2-4 – please present either a box plot or a bar plot for a single dataset, not both. Please provide details of what the different parts of the graph are, e.g. Fig 4B – what do the bars represent – normally this would be the standard deviation is presenting mean data. Why are there fibrinogen levels below 0.0 – how is this possible?
Please remove and replace the term average with the correct statistical term (mean, median or mode as appropriate)
Discussion
I have not reviewed the discussion. The methods and results currently need substantial reworking to meet the STROBE guidelines, including additional methods details, and reviewing the results and statistics. This needs to be done before any discussion of significant findings can be done.
Author contributions – please check and review this - this should just have the correct terms for the contributions along with the initials or the researcher.
References
Comprehensive and detailed reference list.
Very few errors – please just check and correct the formatting of the journal titles, as some have full stops within them and some do not
Comments on the Quality of English Language
In comments above.
Author Response

(The authors gave the same response as above.)

Round 2
Reviewer 2 Report
Comments and Suggestions for Authors
The manuscript in its current version shows evidence of thorough and significant revision, responding substantially to all the comments and suggestions made.
Only a few minor comments remain, which the authors can quickly address without further re-reading by the reviewer.
In the discussion section, it would be interesting to include a final or additional paragraph that highlights the approach taken at the international level for the management and regulation of donkey husbandry and slaughter. This would provide an opportunity for further reflection and potentially serve as an initial model for developing national regulations on these aspects. This paragraph would contribute to strengthening the authors' conclusions.
Line 106: Please check the format of the citation in the text and, if necessary, change the order of the citations accordingly.
Author Response
Muito obrigado por todos os seus comentários e sugestões perspicazes. Sua revisão foi verdadeiramente enriquecedora e contribuiu significativamente para a melhoria do nosso manuscrito. O parágrafo que você sugeriu foi incorporado à seção de discussão (linhas 346–356). Em relação à referência à lei (linha 107), o número da lei aparece entre parênteses dentro do texto, e a citação completa é listada como número de referência 18 (linha 109).
Thank you very much for your opinions and insightful suggestions. His revisions did indeed enrich our manuscript and made a significant contribution to improving it. The paragraph you suggested has been included in the discussion section (lines 346-356). Regarding the citation of lei (line 107), the number of lei appears in parentheses in the main text, and the complete citation is listed as reference number 18(line 109).